# An Approach to Automatic Hard Exudate Detection in Retina Color Images by a Telemedicine System Based on the d-Eye Sensor and Image Processing Algorithms

**DOI:** 10.3390/s19030695

**Published:** 2019-02-08

**Authors:** Emil Saeed, Maciej Szymkowski, Khalid Saeed, Zofia Mariak

**Affiliations:** 1Department of Ophthalmology, Faculty of Medicine, Medical University of Bialystok, 24A Curie-Sklodowskiej Street, 15-276 Bialystok, Poland; emilsaeed1986@gmail.com (E.S.); mariakzo@umb.edu.pl (Z.M.); 2Bialystok University of Technology, Faculty of Computer Science, 45A Wiejska Street, 15-351 Białystok, Poland; m.szymkowski@pb.edu.pl

**Keywords:** hard exudates, retina, d-eye sensor, automatic diagnosis, image processing

## Abstract

Hard exudates are one of the most characteristic and dangerous signs of diabetic retinopathy. They can be marked during the routine ophthalmological examination and seen in color fundus photographs (i.e., using a fundus camera). The purpose of this paper is to introduce an algorithm that can extract pathological changes (i.e., hard exudates) in diabetic retinopathy. This was a retrospective, nonrandomized study. A total of 100 photos were included in the analysis—50 sick and 50 normal eyes. Small lesions in diabetic retinopathy could be automatically diagnosed by the system with an accuracy of 98%. During the experiments, the authors used classical image processing methods such as binarization or median filtration, and data was read from the d-Eye sensor. Sixty-seven patients (39 females and 28 males with ages ranging between 50 and 64) were examined. The results have shown that the proposed solution accuracy level equals 98%. Moreover, the algorithm returns correct classification decisions for high quality images and low quality samples. Furthermore, we consider taking retina photos using mobile phones rather than fundus cameras, which is more practical. The paper presents an innovative approach. The results are introduced and the algorithm is described.

## 1. Introduction

Diabetic retinopathy is one of the commonest reasons for blind registration in the world. Four hundred million people in the world have diabetes, and this number is expected to rise by 2035. 80% of people have retinopathy after 20 years of having the disease [1,2]. It is still a challenge for ophthalmologists, as diabetic retinopathy should be diagnosed before it is symptomatic. Screening examinations are very important but people often forget to see their ophthalmologist until they notice decreased visual acuity. Diabetic retinopathy is a vascular disease. Development of microaneurysms (widening of the retinal vessels) in the capillary network allows plasma and lipids to leak out from the vessels into retina [1]. Those lipids are called hard exudates. They are yellow, shiny flecks in the macular area. Diabetes is the main cause of the hard exudates but they may also be caused by retinal vein occlusion, neuroretinitis, or radiation-induced retinal vasculopathy. Plasma leakage results in retinal oedema. Vision may be significantly reduced. Other lesions may also appear in diabetic retinopathy, such as white cotton-wool spots (as a result of vessel occlusion) or haemorrhages [3]. The earlier and more precisely the disease is diagnosed, the more accurate and successful the first steps taken to deal with the patient’s state will be. Detecting disease in its early stages and providing proper treatment gives a chance of stopping its progression and improving the patient’s visual acuity [4]. The authors’ algorithm could succeed in identifying such cases. Taking a retinal photo using a phone camera may detect small lesions and retinal changes, and suggest whether patients require referral to an ophthalmologist. Treatment depends on visual acuity and the extent of the retinal abnormalities. In some cases, it is enough to observe the patient rather than treating them.

In the literature, different approaches to retina color image processing and hard exudate extraction were found. These solutions can be classified into two groups: the first based on image processing and analysis and the second using artificial intelligence. However, despite the fact that there are multiple diversified approaches connected with retina color image processing, little research has been done in this particular field with high accuracy results. The authors took into consideration only approaches published between 2007 and 2018. Scopus, Web of Science, Springer Link, and Nature databases have been searched. Keywords by which the databases were explored are: retina, hard exudates, pathological changes, ophthalmologic system, telemedicine, and automatic diagnosis. 

The main question is connected with the current state of the art; namely, what techniques were used in currently published algorithms. Moreover, the authors would like to present a few representative examples of both groups of algorithms mentioned above. 

At the beginning we were looking for different approaches to retina color image processing and analysis. The most interesting was [5]. It presented an algorithm to remove the optic disk and vascular pattern from a retina color image before hard exudate detection. The main disadvantage of the proposed method was its time consuming nature and the complexity of its algorithm.

The authors in [6] proposed an interesting approach to retina color image processing towards human recognition. In this work, a fully automated system was presented. The basic algorithm in this approach is scale invariant feature transform (SIFT) used to make retina images invariant of scale and rotation. Moreover, the authors proposed a novel preprocessing method based on the improved circular gabor transform (ICGF). Their approach was fast and was of high accuracy for retina color image processing applied for the sake of user recognition. However, contrary to our approach, it did not deal with pathological changes in the course of human identification.

The second group of algorithms was connected directly with hard exudate detection. The first analyzed paper was based on the Haar wavelet [7]. The algorithm consisted of seven steps in which the authors performed simple operations in the image (like conversion to grayscale or color normalization) as well as more advanced algorithms (like Haar wavelet decomposition and reconstruction). The results presented in [7] show that this approach gave no more than 22.48% accuracy. Hence, the main disadvantage of this approach was definitely its low accuracy level.

Another interesting algorithm is presented in [8]. The authors proposed an approach to hard exudate detection based on morphological feature extraction. Their algorithm consisted of 10 steps, each of them easily classified as either typical or basic image processing methods. Beyond these steps, the authors claimed they needed to implement another algorithm, by which optic disk and bright structures were removed. The main disadvantages of this approach were connected with the high level of complexity and low accuracy in images with different levels of brightness.

An approach using deep learning was presented in [9]. In it, the authors created a deep neural network for hard exudate detection. They used TensorFlow framework for implementation. This solution was time consuming and did not detect small hard exudates in their initial stage.

Literature review has shown there are multiple different techniques used for retina color image processing. We observed simple algorithms based on filtering or morphological operations. There also are approaches that use more advanced techniques, like Haar wavelet decomposition or support vector machine [10] and discrete cosine transform [11]. Moreover, during the initial stage of the research, it was observed that most of the approaches are based on simple image processing methods [12,13,14]. There are also different representatives of the second group of solutions that are soft computing-based. Multiple different techniques were used to detect hard exudates; most of them were connected with neural networks [15,16], genetic algorithms [17], machine learning [18,19], and deep learning [20,21]. They present interesting ideas for hard exudate detection with artificial intelligence methods. However, all of these algorithms are time-consuming, with the disadvantage that small hard exudates in their initial stage are not detected by their solutions.

Another question is connected with the currently presented approaches to telemedicine in ophthalmology. These solutions can be divided into two groups. The first uses intelligent techniques and is fully-automated, while the second needs stationary devices.

The most interesting representatives of the first group are [22,23]. In these approaches, smart algorithms for hard exudate detection were introduced. The second group is represented by [24]. The authors proposed a new approach to telemedicine in ophthalmology, in which a high-resolution fundus camera needs to be used. The authors proposed to use a device placed in a specialized van. When the image is obtained it is sent to ophthalmologist by the internet connection.

In the literature, authors also found approaches to current state-of-the-art review [25,26]. In the case of [26], a comparison between different approaches to smartphone usage in telemedicine was presented. The authors also compared results of different solutions in diabetic retinopathy automatic diagnosis. Neither of these approaches has accuracy higher than 95%.

In this paper, we present our own robust high accuracy (98%) algorithm by which hard exudates can be detected with simple and fast methods. Moreover, description of the fully-automated telemedicine system based on d-Eye sensor is presented. Experiments related to low quality images are also conducted. The results and discussion are described in Section 3 and Section 4, respectively.

## 2. Materials and Methods

### 2.1. Proposed Telemedicine System

Telemedicine is one of the fastest developing branches of science, and combines information technology and medicine. An interesting description of this idea was presented in [27]. We also have contributed to this idea. Our approach is based on novel d-Eye sensor and fully-automated image processing system for pathological change detection in retina color images.

The general architecture of the proposed smart system is presented in Figure 1. It can be divided into two main parts. The first is connected with the end user (patient) and the second consists of image processing, feature extraction and classification algorithms. The software part was implemented with Java programming language (cloud part) and Swift (smartphone part). Moreover, the application was deployed on two cloud platforms: Google Cloud and Microsoft Azure. The authors decided to test these two popular services; both of them allowed results to be achieved in similar times. The cloud part was created as a set of REST (Representational State Transfer) services that are called by the application after the user obtains their retina photo. Each procedure is called using standard HTTP methods (GET, POST, PUT, DELETE). However, to use all services implemented in the cloud, the user has to validate themself. This is done on the basis of third party (Google, Microsoft, GitHub) identity providers and JSON Web Token obtained after successful validation on external pages.

The customer-available components are the smartphone and the d-Eye [28] sensor. In this case, the crucial point of the approach is the d-Eye, a smartphone-based retinal screening system. This sensor can be easily attached to a smartphone. Connection between devices allows the user to change their mobile phone into real ophthalmic camera. After fusion it is easily possible to conduct routine eye examinations and retinal screening. Moreover, the user will get a high resolution retina color image. The d-Eye sensor is composed of two parts: the bumper and the d-Eye lens. This smart sensor uses the camera and the light source provided by the smartphone. The light is redirected from the flash and is projected coaxially to the lens. This allows reflection of the retina image. Moreover, the d-Eye lens eliminates corneal glare, which is a common problem in standard ophthalmoscopes. The d-Eye sensor works properly when it is placed about 1 cm in front of the human eye. Currently, the authors of this device have only prepared overlays for Apple iPhone (from Apple iPhone 5 to Apple iPhone 7), although they are working on versions for other smartphones. 

When a retina color image is obtained by the user, the image processing and analysis procedure starts. At the beginning, the image is sent by remote connection to the cloud-based ophthalmic system. Before this, the data is encrypted using public key cryptography. The next step is connected with image processing, and is fully-described in Section 2.2 of this article. When image processing is finished, an additional process is carried out, dealing with retina vein removal. Finally, the results obtained by the last procedure are applied to the image preprocessed in the first stage. Another step allows removal of the optic disk. After this method, the processing stage is finished. As the last step, stage classification based on image analysis is run. It gives the decision on whether the image contains pathological changes or not. Classification is also described in Section 2.2 of this article.

The proposed system is also ready to use with stationary devices like the Kowa VX-10 Fundus Camera. This means that the physician takes photos with this device and sends them by web portal to the processing system. The diagnosis result is then sent back to their smartphone. In Section 3, we present the results of the proposed approach on the basis of images from different sensing devices, handheld and stationary (i.e., Digital Eye Center Microclear Handheld Ophthalmic Camera HNF and Kowa VX-10 Fundus Camera). The devices used during the research are presented in Figure 2. The parameters of each device are presented in Table 1. All devices were used because the authors would like to apply their solution to a variety of images (with different resolutions and qualities). The results of these experiments are described in Section 3 of this document. 

### 2.2. Methodology

Our approach consists of a few algorithms. The first is used to preprocess the image. With this step we obtain an image without additional superfluous elements that show the retina vascular pattern. The next step gives us the result after removal of the retina vascular pattern and optic disk from the image obtained after the first stage. Finally, we have the classification part, which provides information on whether the eye contains pathological changes or not. The activity diagram of the approach is presented in Figure 3.

#### 2.2.1. Image Preprocessing

The image preprocessing part is mainly connected with our experience acquired during previous research [29,30,31]. The whole algorithm was prepared in Java programming language and was adapted to cloud-based solutions. The block diagram of this algorithm is presented in Figure 4.

The first step is to load the image. It is then converted to grayscale with green channel. In the literature we have found that this channel is normally used in medical image processing [32,33,34]. The experiments conducted by us have also shown that it gives the best results. It means that the image contains the largest number of the details we are looking for. The visual comparison results between different grayscale conversion methods are shown in Figure 5.

The second stage of the preprocessing part of our approach is the histogram stretching. The algorithm is used to increase the contrast of the image [35,36]. This operation is used to cover all possible gray levels (from 0 to 255). New values for each available grayscale level were calculated as in (1).
(1)$$S_{k}^{\prime }=\frac{S_{k}-k_{min}}{k_{max}-k_{min}}\cdot Z_{k}$$
where $S_{k}^{\prime }$ is the new value for *k*-th level, $S_{k}$ is the original value for *k*-th level, $k_{min}$ and $k_{max}$ are minimum and maximum level, respectively, in the original histogram, whilst $Z_{k}$ is number of new possible ranges. In this case, $Z_{k}=256$ because we use all values from 0 to 255. The result obtained with this operation is presented in Figure 6.

The next step is connected with noise removal from the image. We deal with this task using a median filtering operation. The result is presented in Figure 7.

The last step of the image preprocessing stage is gamma correction. Retina color images can be represented in different levels of brightness (from bright red to even blood red). This fact requires us to unify the color of the image. This was done with a gamma correction method, described in [37]. The gamma parameter value was calculated as in (2). Each pixel was multiplied by the gamma value. The result of this step is presented in Figure 8.
(2)$$\gamma =\frac{-0.3}{log_{10}X}$$
where *X* is the mean pixel value calculated on the basis of all pixels in the image.

This step ends the image preprocessing stage. The image is now moved to another module that extracts the vascular pattern. Another stage will concern vascular pattern and optic disk removal. 

#### 2.2.2. Retina Vascular Pattern Extraction

The second module of the proposed approach is needed to extract the retina vascular pattern. During discussion with ophthalmologists, it was pointed out that it is not possible to observe hard exudates on vascular patterns. This concludes that information about vascular patterns is not useful for detection of pathological changes, meaning that we can remove such information from our image. The block diagram of the proposed algorithm for retina vascular pattern extraction is presented in Figure 9.

The first step is once again conversion of the image to grayscale on the basis of green channel value. As was presented in Section 2.2.1, this value is used because it guarantees the most precise results and that the number of the details in the image is the highest. 

Another step, noise removal, is also done with median filtering. This algorithm is an efficient way to get precise results. Original image and images obtained after both of these steps are presented in Figure 10.

The third and the fourth steps are connected with image enhancement before vessel segmentation. It is done on the basis of histogram equalization (which allows enhancement of the image contrast) and brightness correction. Images obtained after both of these stages are presented in Figure 11.

The main aim of this algorithm is vascular pattern extraction. The next procedure allows segmentation of vessels from the retina image. This is done with a Gaussian matched filter [38]. All twelve masks were used to detect vessels in the retina image. The result is presented in Figure 12.

To obtain proper vascular patterns, a local entropy binarization algorithm was used to get proper white veins on a black background. However, this has caused a few small elements that do not belong to retinal veins to be shown in the image, alongside the marked veins. Additional elements are removed on the basis of their length. If they are too short (the length in pixel number is selected arbitrarily), they are deleted–we assume that these elements are not parts of real veins. The results of both of these steps are presented in Figure 13.

Via this step, the vascular pattern of the processed retina was obtained. In the form presented in Figure 13b, it will be applied to the obtained image after the preprocessing stage (Figure 8b). Both of these images are considered in the next module.

#### 2.2.3. Removal of Retina Vascular Pattern and Optic Disk

This module is responsible for image preparation through to final classification. The first step of the algorithm is to remove the vascular pattern that was extracted in the second stage from the final image in the first step. Removal means that all pixels belonging to vessels will be marked in black color. The image obtained after vascular pattern removal is presented in Figure 14.

The following stage is image binarization. Via this step, pathological changes can be extracted. The authors tested a few different automatic binarization methods and the experiments showed that the best results were obtained when the binarization threshold was set to 13. We observed that pathological changes were shown as well as the optic disk. The results after binarization are presented in Figure 15. The last step of this procedure is optic disk removal. The ophthalmologist opinion is that there is no possibility of hard exudates occurring on the optic disk, and hence it can be removed.

The optic disk is removed on the basis of data entered by the user regarding whether they are dealing with the left or right eye (these differ in terms of the optic disk position in the retina color image). In our program, the user (patient) sets this information at the beginning of the diagnostic procedure. Before removal, we calculate image variance of the last binarized sample. It allows observation of which parts of the image are characterized by the greatest variability. This image is presented in Figure 16.

After image variance calculation we can remove the optic disk. We are looking for the first white pixel from the side (left or right eye) selected by the user. When this pixel is found, we remove the 80 × 80 square [39] by marking all of these pixels in black color. The image after optic disk removal is presented in Figure 17.

#### 2.2.4. Classification

The last part of our approach is connected to classification. As the first step, the image obtained from the third module is applied to the original retina color image. All changes are marked in blue color. The resulting image is presented in Figure 18. The classification algorithm is simple; if blue points are visible in the image obtained after optic disk removal in retina color image, we can conclude that the image contains pathological changes. In the other case (i.e., no blue points in retina color image), the retina is healthy.

## 3. Results

The first experimental part was conducted on a database consisting of 100 samples (50 healthy retinas and 50 with pathological changes). All of them were obtained with Kowa VX-10. The database was created on the basis of samples from 67 patients (some of them are represented by two samples taken at different times–the first at the beginning of treatment and the second one after a certain time from the treatment beginning). All samples were acquired during medical examinations at Białystok University Clinical Hospital. There, we used the whole presented method with classification part. As was mentioned in the second chapter, the decision was made on the basis of blue points that could be observed in retina color image after applying the results of our approach. The experiment showed that the proposed approach had an accuracy level of 98%. It was observed that only two healthy images were evaluated as images with pathological changes. 

For the first experiment the authors answered the following question: Does the proposed system correctly classify images with pathological changes? It was necessary to measure the values of two parameters: false acceptance rate (FAR) and false rejection rate (FRR). On the basis of the conducted experiment, the conclusion was that FAR parameter level was 2%. False acceptance means recognizing a retina color image as a sample with pathological changes when it represented a healthy eye; that is, the system would incorrectly recognize pathological lesions in healthy eyes. Moreover, we checked the value of the FRR parameter. False rejection is a situation when retina color image contains pathological changes but in fact it is recognized as a healthy eye. FRR for our algorithm was equal to 0. 

The first author, an ophthalmologist, pointed out that it is better to have higher FAR than FRR because patients with diabetes can never have too many eye fundus examinations, especially when the duration of their disease is significantly long. The duration of the diabetes mellitus is the most valid risk factor for development of diabetic retinopathy.

The second experiment was connected to checking whether our algorithm could be used for different images—ones with high resolution and high quality, and ones obtained by devices with worse parameters. Our database, obtained by the lower quality devices, contained 60 samples (50 healthy retinas and 10 with pathological changes). Thirty healthy samples were obtained with Digital Eye Center Microclear Handheld Ophthalmic Camera HNF whilst the rest of the low quality images were acquired with the d-Eye sensor. All of samples were acquired during medical examinations at Białystok University Clinical Hospital. Comparison was done to check if it was possible to observe pathological changes in retina images from worse devices. The sample image is presented in Figure 19.

The results showed that in the case of retina images from devices with lower precision, none of the healthy pictures was classified as a sample with pathological changes. Moreover, all retinas with diabetic retinopathy were classified as having pathological changes. This experiment confirmed that our solution can also be easily used with lower quality images. Moreover, it was pointed out that no additional adjustment of the proposed approach was needed.

It should also be pointed out that all samples (high quality and low quality) were obtained in a clinical setting. This allowed us to obtain retina images taken with the highest precision by an experienced ophthalmologist. High quality samples were taken by the device with the highest possible resolution (Kowa VX-10) whilst low quality images were obtained with the other two devices.

The summary of the results obtained for both experiments is shown in Table 2.

The authors calculated values for parameters FAR and FRR as well as for sensitivity and specificity. Statistical classification functions were calculated as in (3) and (4). Obtained results are presented in Table 3.
(3)$$Sensitivity=\frac{\left|True\;Positivite\;samples\right|}{\left|True\;Positive\;samples\right|+\left|False\;Negative\;samples\right|}$$
(4)$$Specificity=\frac{\left|True\;Negative\;samples\right|}{\left|True\;Negative\;samples\right|+\left|False\;Positive\;samples\right|}$$

## 4. Discussion

The purpose of the study was to create and implement an algorithm which helps diabetic patients examine their eye fundus. Our method can extract pathological lesions such as hard exudates in diabetic retinopathy and separate them from healthy eyes. The algorithm is introduced to color fundus retina photos from patients of Medical University Clinical Hospital. We take into consideration diabetes mellitus, as it is one of the commonest and most dangerous diseases. Little research has been done by other researchers so far in this particular field. 

During our research we were mainly looking for the answers to four questions. These were:

Would it be possible to create a fast and precise algorithm for hard exudate detection in retina color images?

Can we implement an algorithm that can be used in high quality images in low quality samples without precision reduction?

Should we expect higher false acceptance rate (FAR) than false rejection rate (FRR) in the case of pathological change detection in retina color images?

Can we use simple overlay on the smartphone for retina color image acquisition, and is it possible to implement a fully automated diagnostic system?

The conducted experiments provided the answer to all questions. The authors had various meetings with other experienced ophthalmologists. We asked them if it is better to detect some false changes in healthy retina images than to miss some little changes in the images with pathological lesions. The answer was that we should prepare a solution that could detect some false changes in the healthy samples rather than the second case. This was connected to the fact that it is better to inform the patient about possible pathological changes than to miss them and send information that eye is healthy.

In the case of the first question, it has to be pointed out that our solution has 98% accuracy. On a 100-element database, only two healthy images were classified as samples with pathological changes. Moreover, we measured the time needed to obtain the results. The first measurement was done only for the whole algorithm (without smartphone and web communication). The time after which the decision was presented was equal to 6 seconds. This test was run on a personal computer with one Intel Core i7 CPU, 16 GB RAM and 256 GB SSD hard drive. The second measurement was done for the whole system. The authors took into consideration communication between the smartphone, the d-Eye sensor, and the cloud-based program. As was mentioned in the description of the system architecture, the sensor was an overlay that used smartphone flash (so here we do not observe any delay), although the smartphone program calls REST services with standard HTTP methods and the data is encrypted before sending over the network (so here we can note some delays). Moreover, the user has to validate themself before they can use the cloud-based software. The examination results were obtained after 1 min and 15 s. For this case, we concluded that the web connection, data encryption, and communication procedures can be reasons for the additional time required to finish the procedure.

The second question was resolved by another experiment. As was mentioned in the third chapter, we also used low quality images for the pathological change recognition. We had healthy retina samples as well as diabetic retinopathy images. The results showed that our solution could be used in low quality images without any additional adjustment. During the experiments the authors did not observe any reduction in the algorithm accuracy.

The last question was in fact the most crucial and basic one. The authors implemented a fully automated pathological change detection algorithm and the whole cloud-based application together with the program for an iOS operating system. The proposed solution is easy to use with the d-Eye sensor and can be a real help for every person who would like to check the state of their health, not necessarily just those suffering from diabetes.

Another experiment that was done by the authors was a comparison with other solutions. We took into consideration the algorithms’ accuracies, sensitivities, and specificities. The results of our experiment have shown that the approach proposed in this article is competitive in comparison with the other solutions described in various research papers. A comparison of the proposed solution with others is presented in Table 4. 

In Table 5, the authors present a summary of the techniques used in this paper, and others to which they are compared.

To sum up this section, the proposed cloud-based solution can be used by everyone because it is simple and its usage allows for the user to visit the ophthalmologist with initial examination results. The patients do not need to wait for examinations; they can do them on their own. Moreover, in comparison with the other researchers’ solutions, the proposed approach has higher accuracy level. Also, no images were mistakenly rejected (even if they had pathological changes).

## 5. Conclusions

The proposed algorithm detected hard exudates with an accuracy of 98%, which is significantly more precise than known ones. This may help patients with diabetic retinopathy get to an ophthalmologist before symptoms occur. Moreover, the solution is characterized by a high accuracy level and also short time of operation to get the precise examination results. 

The approach presented in this paper was implemented in real development environment and was tested for accuracy and for data safety. We used different encryption algorithms to make data safer. 

In the future, the authors would like to create a fully automated eye diagnostic system that can be used for detection of much more different diseases. We will take into consideration diseases other than diabetic retinopathy.

The authors’ current work is to improve the proposed algorithm with soft computing methods for accuracy results even higher than 98%. Moreover, we are working on implementing the hardware of our own sensor device for retina color image acquisition.

## Figures and Tables

**Figure 1 sensors-19-00695-f001:**
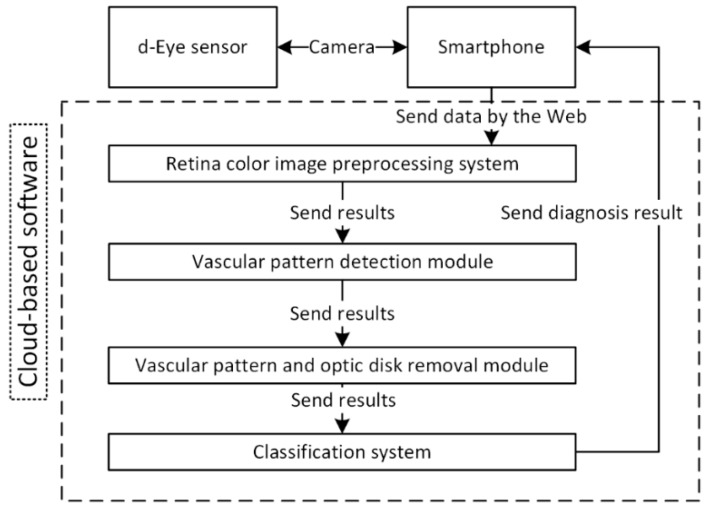
The architecture of the proposed system.

**Figure 2 sensors-19-00695-f002:**
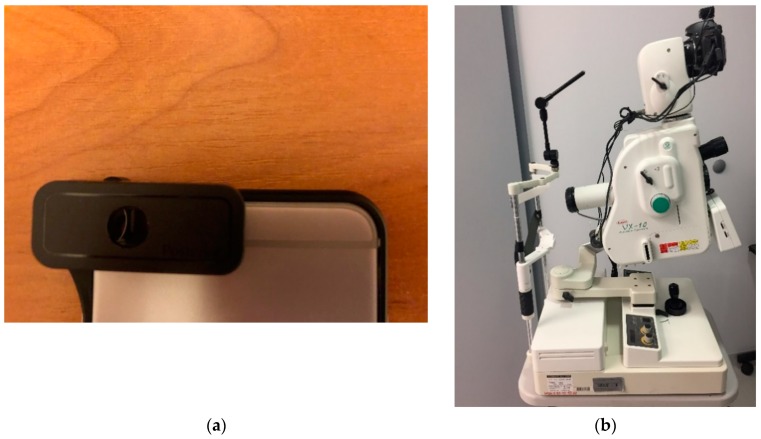
All devices used during research: (**a**) d-Eye overlay for smartphone, (**b**) Kowa VX-10 Fundus Camera, and (**c**) Digital Eye Center Microclear Handheld Ophthalmic Camera HNF.

**Figure 3 sensors-19-00695-f003:**
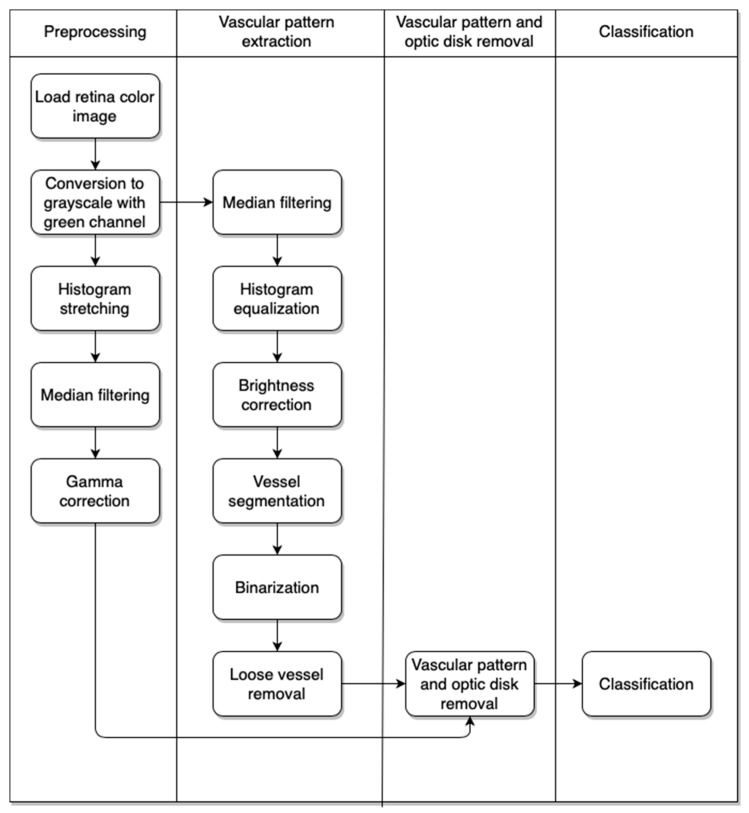
The activity diagram of the proposed approach.

**Figure 4 sensors-19-00695-f004:**
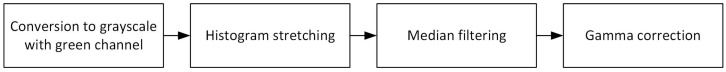
The block diagram of the image preprocessing algorithm.

**Figure 5 sensors-19-00695-f005:**
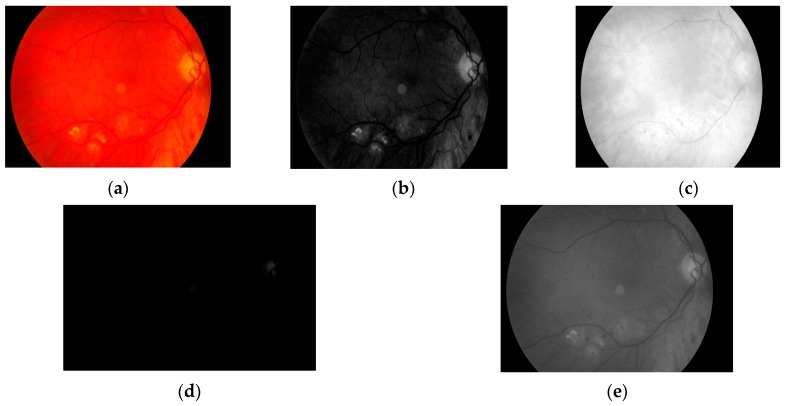
Visual comparison between different grayscale conversion methods: (**a**) original image; (**b**) green channel; (**c**) red channel; (**d**) blue channel; (**e**) average value of all channels.

**Figure 6 sensors-19-00695-f006:**
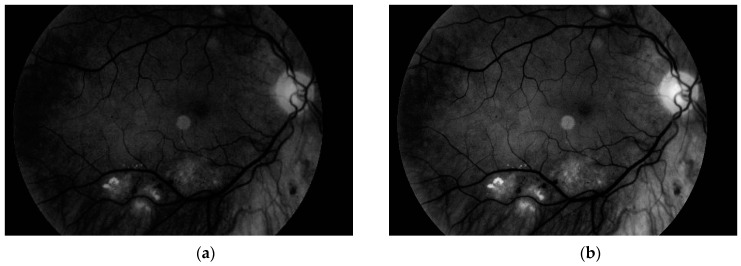
(**a**) Grayscale image and (**b**) its form after histogram stretching.

**Figure 7 sensors-19-00695-f007:**
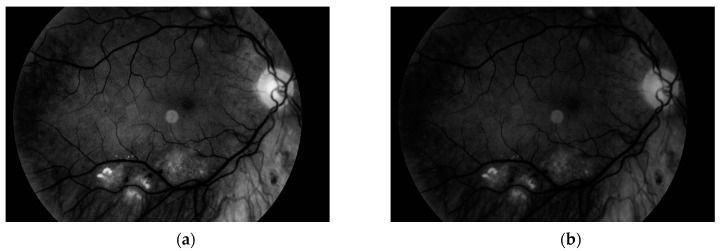
Image after (**a**) histogram stretching and (**b**) median filtering.

**Figure 8 sensors-19-00695-f008:**
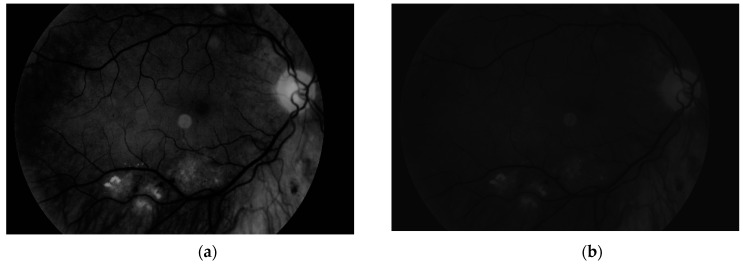
Image (**a**) after median filtering and (**b**) after gamma correction.

**Figure 9 sensors-19-00695-f009:**
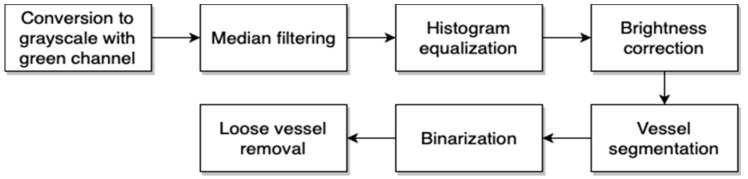
The block diagram of the proposed retina vascular pattern extraction.

**Figure 10 sensors-19-00695-f010:**
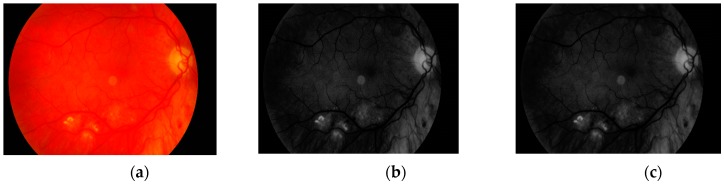
(**a**) Original image, (**b**) image after conversion to grayscale with green channel and (**c**) image after noise removal.

**Figure 11 sensors-19-00695-f011:**
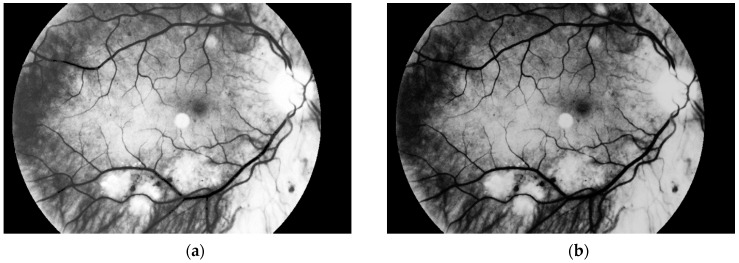
Image after (**a**) histogram equalization and (**b**) after brightness correction.

**Figure 12 sensors-19-00695-f012:**
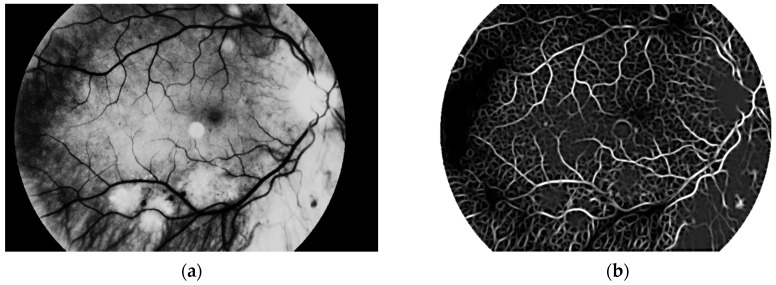
Image after (**a**) brightness correction and (**b**) Gaussian matched filter.

**Figure 13 sensors-19-00695-f013:**
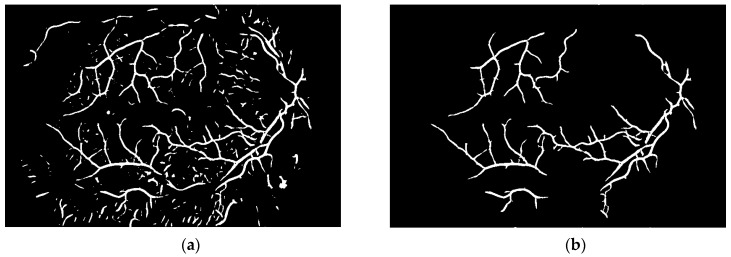
Image (**a**) after binarization and (**b**) after short vessel removal.

**Figure 14 sensors-19-00695-f014:**
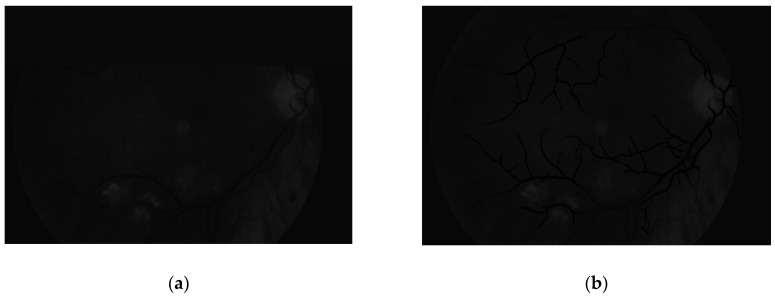
Image (**a**) after the first step and (**b**) after vascular pattern removal.

**Figure 15 sensors-19-00695-f015:**
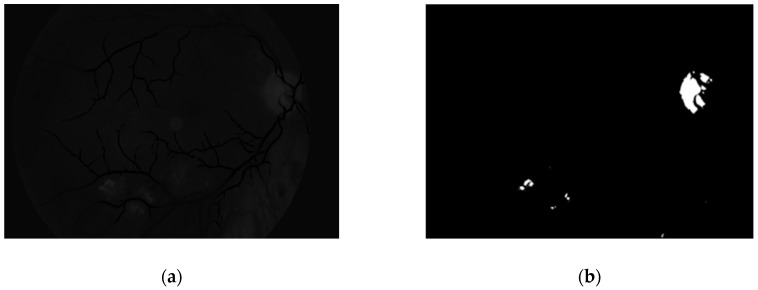
Image (**a**) after vascular pattern removal and (**b**) after binarization.

**Figure 16 sensors-19-00695-f016:**
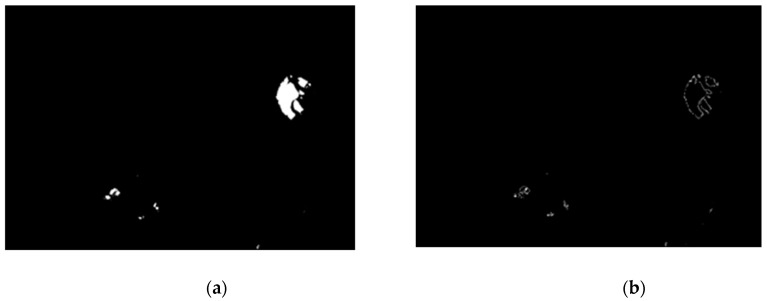
Image (**a**) after binarization and (**b**) calculated image variance.

**Figure 17 sensors-19-00695-f017:**
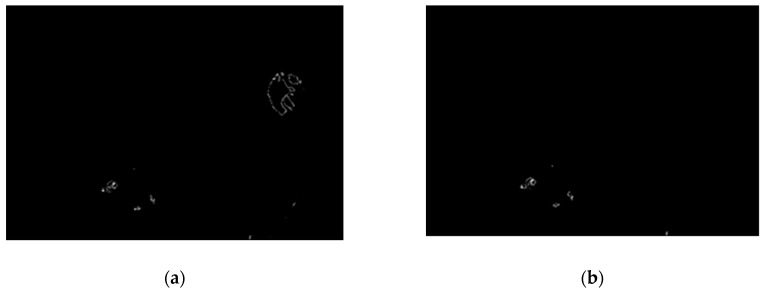
Image after (**a**) calculating its variance and (**b**) after optic disk removal.

**Figure 18 sensors-19-00695-f018:**
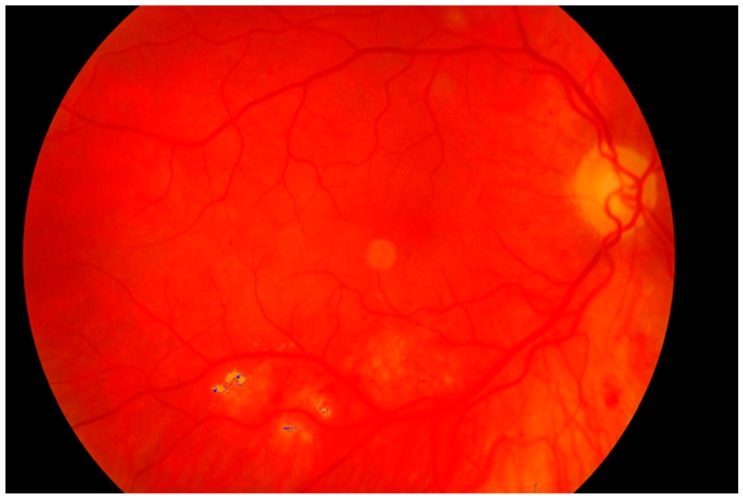
Retina color image with marked pathological changes.

**Figure 19 sensors-19-00695-f019:**
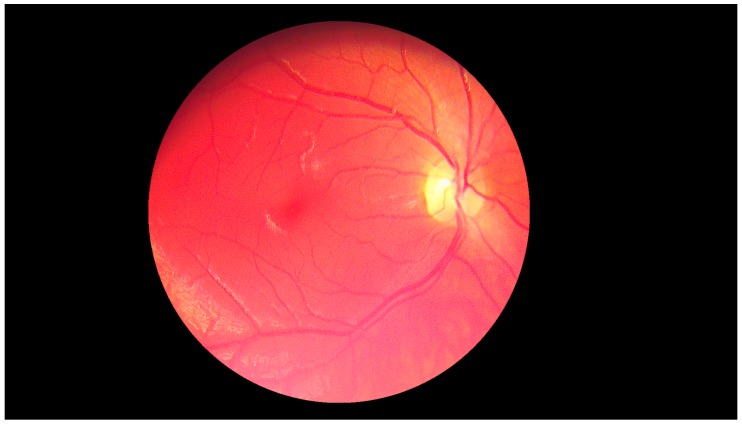
Retina color image obtained by the device with worse parameters. No pathological changes exist.

**Table 1 sensors-19-00695-t001:** Comparison of all device parameters.

Device Name	Price	Quality (in Comparison to the Best)	Weight	Refractive Compensation	Image Resolution
Kowa VX-10	40,000 EUR	High quality	35.5 kg	−32D~+35D	3000 × 2008
Digital Eye Center Microclear Handheld Ophthalmic Camera HNF	5000 EUR	Low quality	0.45 kg	−20D~+20D	1920 × 1080
d-Eye Sensor	400 EUR	Low quality	0.1 kg	−10D~+8D	864 × 1536

**Table 2 sensors-19-00695-t002:** The summary of the conducted experiments.

Type of Images	Source of the Images	Number of Samples	Classification as Healthy Retina	Classification as Sample with Pathological Changes
High quality healthy retina images	Kowa VX-10	50	48	2
High quality retina images with pathological changes	Kowa VX-10	50	0	50
Low quality healthy retina images	Digital Eye Center Microclear Handheld Ophthalmic Camera HNF, d-Eye Sensor	50	50	0
Low quality retina images with pathological changes	d-Eye Sensor	10	0	10

**Table 3 sensors-19-00695-t003:** Measured parameters of the proposed algorithm.

Parameter	FAR	FRR	Sensitivity	Specificity	Accuracy
Value	**2%**	**0%**	**100%**	**96%**	**98%**

**Table 4 sensors-19-00695-t004:** Comparison of the proposed solution with different ones described in other works.

Algorithm	Accuracy	Sensitivity	Specificity
**Joshi, Karlue [8]**	**91%**	**96.7%**	**85.4%**
**Benzamin, Chakraborty [9]**	**98.6%**	**98.29%**	**41.35%**
**Garcia, Sanchez et al. [15]**	**97.01%**	**100%**	**92.59%**
**Khojasteh, Aliahmad, Kumar [21]**	**98%**	**96%**	**98%**
Our proposed algorithm	98%	100%	96%

**Table 5 sensors-19-00695-t005:** Summary of the used techniques.

Algorithm	Techniques
Joshi, Karlue [8]	Image processing techniques that are mainly based on morphological feature extraction.
Benzamin, Chakraborty [9]	Deep learning techniques for hard exudate detection implemented with TensorFlow framework.
Garcia, Sanchez et al. [15]	An approach that uses neural network: multilayer perceptron (MLP), radial basis function (RBF), and support vector machine (SVM).
Khojasteh, Aliahmad, Kumar [21]	Convolutional neural network was trained to detect hard exudates in retina color images.
Our proposed algorithm	Image processing techniques, veins, and optic disk removal before hard exudate detection.

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
