# Peer review of "An Approach to Automatic Hard Exudate Detection in Retina Color Images by a Telemedicine System Based on the d-Eye Sensor and Image Processing Algorithms"

_sensors, 2019, doi:10.3390/s19030695_

Round 1
Reviewer 1 Report
This is a topic which describes an algorithm that extracts pathological changes in diabetic retinopathy – hard exudates.
The authors used classical image processing methods, a fully automated pathological changes detection algorithm, a cloud-based application together with the program for Android Operating System and the results have shown that the proposed solution accuracy level equals 98%.
This is not a novel study as image extract studies has been previously reported in ophthalmology. Authors noted strengths and limitations of the study.
I found the article to be well written report on a topic of interest to telehealth/ teleophthalmology academia.
Minor corrections of English may be needed.
Line 363 “To sum up this chapter”- this is not a thesis chapter!
Frequent use of “not only … but also’ need to be avoided.

Author Response
Thank you very much for your opinion and valuable comments. Our answers to your inquiries and our changes are given in the following table.
Reviewer comment | Authors’ answer and changes |
Line 363 “To sum up this chapter” – this is not a thesis chapter! | We are thankful for your comment. You are of course right, we changed this sentence to “To sum up this section…”. |
Frequent use of “not only … but also” need to be avoided. | Thank you for your valuable comment. We made a language inspection of our paper. In most of the cases we changed “not only” and “but also” to other different language forms. |
Reviewer 2 Report
Digital image processing of fundus images for the detection of signals of diabetic retinopathy has been an active research field for more than 10 years. The use of such a technique in the contexct of a Telemedicine setup for the remote triage of patients has also been performed for some time now (se, e.g. Kanagasingam).
This work suggests a new image processing pipeline aimed specifically at the detection of hard exudates. Differently from the present research tendency for image processing-based patient triage, focused on deep learning CNNs, this proposal is based upon a set of traditional image processing algorithms. This has the advantage of simplicity, has also the potential disadvantage of lack of robustness, which the study takes into consideration through the use of 50 "low quality images" in the validation study.
The results presented, even if performed with a reduced dataset, seem to be, with 98%, much better than the average results found in the literature.
However, there are various methodological problems with this article:
The literature review was performed in an ad-hoc manner, with the authors discussing a few selected papers and comparing them to their approach. Since this is a well-established research field, that has been around for some time, I would expect a more formal, systematic literature review;
The authors do not systematically compare their approach with existing approaches in the literature. For the same reasons as above, I would expect (a) a table comparing the techniques used in their approach with techniques in other approaches and (b) a table comparing the accuracy (including false positives and false negatives) of these approaches;
The authors claim that they have employed three different acquisition devices in order to capture the 150 images used in this study: d-Eye overlay for smartphone (a), Kova VX-10 Fundus 131 Camera (b) and Digital Eye Center Microclear Handheld Ophthalmic Camera HNF (c). However, thy do not cite anywhere, how many of these images where gained with each of these devices, they only cite that they employed 50 "low quality healthy retina images". They should explicit the source of the images.
And where are the "low quality retina images with pathological changes"? Only high quality images present pathologies. The lack of one diagnostic category of images in a one of the image sets is also a flaw.
Author Response
We are really thankful for your valuable comments. We prepared the following table that summaries our answers to your inquiries and our changes.
Reviewer comment | Authors’ answer and changes |
The literature review was performed in an ad-hoc manner, with the authors discussing a few selected papers and comparing them to their approach. Since this is a well-established research field, that has been around for some time, I would expect to a more formal, systematic literature review. | Thank you for your comment. We changed literature review. At the beginning we present two, most important general approaches (the one that is based on image processing and the second artificial intelligence based). Then we present the most interesting algorithms of each group. Both of them were also summarized. Another improvement was additional literature review about telemedicine. We present two groups of this idea – stationary and mobile. Moreover, we present a few interesting approaches connected with both of them. |
The authors do not systematically compare their approach with existing approaches in the literature. For the same reasons as above, I would expect (a) a table comparing techniques used in their approach with techniques in other approaches and (b) a table comparing the accuracy (including positives and false negatives) of these approaches. | We are thankful for your comment. We added two tables (Table 3 and Table 4) in section number 4 (Discussion). The first of them presents a comparison between our approach and other approaches (we take into consideration – accuracy, sensitivity and specificity). The second table presents summary of the used techniques. In both of the cases we chose four most interesting approaches (to which our idea was compared). |
The authors claim that they have employed three different acquisition devices in order to capture the 150 images used in this study: d-Eye overlay for smartphone (a) Kowa VX-10 Fundus 131 Camera (b) and Digitial Eye Center Microclear Handheld Ophthalmic Camera HNF (c). However, they do not cite anywhere, how many of these images were gained with each of these devices, they only cite that they employed 50 “low quality healthy retina images”. They should explicit the source of the images. | Thank you for your valuable comment. We described each of three used devices in Table number 1 (section 2.1). We added information about: price, quality, weight, refractive compensation and image resolution. Moreover, we added information about how many images were gained with each device (and with which quality) in section 3. We also provided information that all images were obtained in medical conditions. We also pointed out that we used all these devices because we would like to check whether it will be possible to properly mark pathological changes in images from different devices. |
And where are the “low quality retina images with pathological changes”? Only high quality images present pathologies. The lack of one diagnostic category of images is a one of the image sets is also a flaw. | Thank you very much, indeed. Right now, we have 10 samples with pathological changes that were obtained with low quality devices. We added this information in the table. Moreover, the experiments have shown that for new 10 samples, pathological changes were properly detected. |
Reviewer 3 Report
Manuscript Number: sensors-417611
Title:
An Approach to Automatic Hard Exudates Detection in Retina Color Images by Telemedicine System Based on d-Eye Sensor and Image Processing Algorithms
Emil Saeed, Maciej Szymkowski, Khalid Saeed , Zofia Mariak
The topic under discussion is interesting. The authors show results the algorithm that can extract pathological changes in diabetic retinopathy - hard exudates. The text feels more divulgative than technical, since it leaves many undeveloped questions. The authors own some articles on this field with a more detailed description of the software and its implementation.
With the purpose of improving the current document the following changes are suggested:
· Authors describe in the abstract the data handled and presented in the results "68 patients - 39 females and 28 males with an age ranging between 50 and 64 were examined", but there's no explanation about the selection procedure in methodology section. In the results a hundred images are mentioned (50 with pathology and 50 without it). This question should be cleared.
· There's no detailed description of d-eye, only a reference, even though it was described in articles 21 and 22. I believe a short description would fit the article better than just a reference. The camera, sensor and algorithm are the main key of the article but none of them are well described.
· In the methodology the pre-processing and processing methodology is mentioned, but nothing about software implementation (language, app or cloud based) is said. Everything seems blurry. It should be considered to add this information so readers may picture how the system works physically in a more precise way.
· At the start three systems of image capture are discussed: d-Eye overlay for smartphone (a), Kova VX-10 Fundus 131 Camera (b) and Digital Eye Center Microclear Handhel Opthalmic Camera HNF (c), but no further information is exposed. Adding extra information like a price range, quality of the images captured, etc. could improve the section.
· In results section a hundred images' result is presented but there's no information about how they were captured and with which system, neither it is mentioned the criteria used to differentiate between high- and low-quality images. It's all quite blurry. I do not understand either why the three capturing system have been used to compare results. It is not enough saying that low quality images may induce false positives, detecting pathology where there isn't. I think it's way interesting to mention when its detected in images that has. Overall this whole section is poor in its presentation and quite imprecise, which make for a conclusion that don't support the data. Authors should perform more experiments and present them appropriately, so their conclusions are backed up by the data.
· The discussion section is more of a reflection by the authors, even when presenting metrics of time of diagnostic must be acknowledged. Still, for that to be done properly, more information about the communications system and the way the software is executed should be presented. It is important to perform a detailed discussion in this section, where authors compare their results with others achieved by other authors. That way readers may be able to build a better picture of the findings revealed by this paper. It may also help to reinforce the conclusions.
· References should be updated, only 14 or 32 range from 2015 to 2019. There are many works in the same line since the topic researched is of current interest.
· After passing the document through the anti-plagiarism software, the percentage is shown as 9%. I see that this doesn´t affect the content of the document. I will attach the file.

Author Response
Thank you very much for your opinion and valuable comments. Our answers to your inquires and our changes are presented in the following table.
Reviewer comment | Authors’ answers and changes |
Authors describe in the abstract the data handled and presented in the results “68 patients – 39 females and 28 males with an age ranging between 50 and 64 were examined” but there’s no explanation about the selection procedure in methodology section. In the results a hundred images are mentioned (50 with pathology and 50 without it). This question should be cleared. | Thank you very much for your valuable comment. In the paper we added information that we collected 100 images from 67 patients because some of them were represented by two samples (one that was obtained at the beginning of treatment and the second one after some time of treatment). Moreover, we added information that all images were obtained in medical conditions. |
There’s no detailed description of d-eye, only a reference, even though it was described in articles 21 and 22. I believe a short description would fit the article between than just a reference. The camera, sensor and algorithm are the main key of the article but none of them are well described. | We are thankful for your comment. We added information about d-Eye (especially how the retina image is projected and what the conditions are for the best quality of images and how it has to be used). Moreover, we provided a short description about the problem that is solved by d-Eye in comparison to standard ophthalmoscopes (corneal glare). |
In the methodology the pre-processing and processing methodology is mentioned, but nothing about software implementation (language, app or cloud based) is said. Everything seems blurry. It should be considered to add this information so readers may picture how the system works physically in a more precise way. | Thank you for your comment. We added information about languages used for implementation. Moreover, we presented how the system was composed (REST Services) and we also provided information about two cloud platforms on which our application was deployed. Additional information about user identification was also described. This information is presented in section 2.1. |
At the start three systems of image capture are discussed: d-Eye overlay for smartphone (a), Kowa VX-10 Fundus 131 Camera (b) and Digital Eye Center Microclear Handheld Ophthalmic Camera HNF (c), but no further information is exposed. Adding extra information like a price range, quality of the images captured etc. could improve the section. | Thank you very much for your comment. We provided additional information about each device: price, quality, weight, refractive compensation and image resolution. All of them are presented in Table number 1 (in section 2.1). |
In results section a hundred images’ result is presented but there’s no information about how they were captured and with which system, neither it is mentioned the criteria used to differentiate between high- and low-quality images. It’s all quite blurry. I do not understand either why the three capturing system have been used to compare results. It is not enough saying that low quality images may induce false positives, detecting pathology where there isn’t. I think it’s way interesting to mention when its detected in images that has. Overall this whole section is poor in its presentation and quite imprecise, which make for a conclusion that don’t support the data. Authors should perform more experiments and present them appropriately, so their conclusions are backed up by the data. | We are very thankful for your valuable comment. At the beginning we have added information that we used three devices because we would like to check whether it will be possible to observe pathological changes also in low-quality images. We also described in which conditions each of them was obtained and which devices were used to obtain each group. We also provided this information in Table 2 which was added only for this purpose. Another improvement was connected with the analysis of low-quality images of pathological changes. Moreover, we provided a comparison between our approach (in terms of accuracy, specificity and sensitivity) to other algorithms. We also made a short comparison between our idea (methodology) and other solutions. This information is provided in the new Table 3 and Table 4 (section 4) prepare in accordance with your recommendations. |
The discussion section is more of a reflection by the authors, even when presenting metrics of time of diagnostic must be acknowledged. Still, for that to be done properly, more information about the communications system and the way the software is executed should be presented. It is important to perform a detailed discussion in this section, where authors compare their results with others achieved by other authors. That way readers may be able to build a better picture of the findings revealed by this paper. It may also help to reinforce the conclusions. | Thank you for your comment. We added information in section 2.1 and 4 about communications system and the way in which software is executed. Moreover, we presented comparison between our approach and four selected algorithms (in terms of accuracy, sensitivity and specificity). We also made a short comparison connected with used techniques in our idea and four selected. |
References should be updated, only 14 or 32 range from 2015 to 2019. There are many works in the same line since the topic researched is of current interest. | Thank you for your comment. We changed a few source papers and added some new articles (published between 2015 and 2018). |
After passing the document through the anti-plagiarism software, the percentage is shown as 9%. I see that this doesn’t affect the content of the document. I will attach the file. | Thank you very much for this information. We have analyzed the report you attached and indeed we are happy this would not affect the content of the document. |
Round 2
Reviewer 2 Report
On line 84 the authors say that "Literature review has shown there are multiple different techniques used for retina color image processing.". However, they do not describe how this review was performed, which timeframe was taken into consideration, which search strings and which databases were used and which were the results. In this light, the literature review presented in this paper is still ad hoc and not systematic.
The authors are from the medical domain and should be familiar with systematic literature reviews. For the field of Computer Sciences, we suggest the use of:
@article{kitchenham2007,
title={Guidelines for performing Systematic Literature Reviews in Software Engineering},
author={Kitchenham, Barbara},
journal={Keele University, EBSE Technical Report EBSE-2007-01},
pages={1--65},
year={2007}
}
Author Response
Thank you very much for your comment. We have added description of the used literature in our work. Please see the attached new version. Thank you once again for your valuable notes.

Reviewer 3 Report
After applying the suggested corrections the article is more precise, with the conclussions being supported by the data presented. Thus the information exposed has improved significantly.
Author Response
Thank you for reminding us to improve the description of results. We have now added a table with a text to explain more deeply. We hope it will satisfy your expectations. Thank you very much.

This manuscript is a resubmission of an earlier submission. The following is a list of the peer review reports and author responses from that submission.